# Towards Faster Quantum Circuit Simulation Using Graph Decompositions, GNNs and Reinforcement Learning

**Alexander Koziell-Pipe**[*]    **Richie Yeung**[*]    **Matthew Sutcliffe**[*]
Department of Computer Science
University of Oxford
Oxford, UK
`[firstname.lastname]@cs.ox.ac.uk`

## Abstract

In this work, we train a graph neural network with reinforcement learning to more efficiently simulate quantum circuits using the ZX-calculus. Our experiments show a marked improvement in simulation efficiency using the trained model over existing methods that do not incorporate AI. In this way, we demonstrate a machine learning model that can reason effectively within a mathematical framework such that it enhances scientific research in the important domain of quantum computing.

In the present-day 'Noisy Intermediate Scale' (NISQ) era of quantum computing [30], quantum resources are still largely limited. Given this limit on quantum resources, being able to simulate quantum computations efficiently and at scale on classical hardware can accelerate quantum computing research and set a standard for benchmarking quantum computers.

While in general quantum circuit simulation can be #P-hard [28], a subclass of quantum circuits known as stabiliser circuits can be simulated in polynomial time with respect to size [1]. Hence a technique for simulating quantum circuits is to decompose them into an ensemble of efficiently simulated stabiliser circuits, the aggregation of which simulates the same computation as the original circuit. Decompositions are calculated iteratively, where sub-circuits are decomposed in a sequential manner until the ensemble of stabiliser circuits is achieved. At each step in the decomposition, the choice of sub-circuit to decompose can greatly affect the number of stabiliser circuits that need to be simulated at the end – in the worst case, this is exponential with respect to the number of a certain type of gate, called a $T$-gate, in the original circuit.

In this work, we formulate the challenge of choosing good sub-circuit decompositions as a reinforcement learning problem, where an agent learns to make decisions in a combinatorially large action space. To facilitate this, we utilise a mathematical framework known as the ZX-calculus, in which quantum circuits are represented as graphs and reasoning amounts to a set of rules allowing one graph to be transformed into another. Formulating our problem in terms of graphs enables the use of Graph Neural Networks (GNNs), which have seen promising applications in other scientific domains including bioinformatics [39], social networks [17], and combinatorial optimisation [7].

We show that, for classes of quantum circuit known not to be efficiently classically simulated, our GNN agent trained using reinforcement learning achieves significantly more efficient decompositions compared to current methods that do not incorporate AI. Moreover, we show that additional algebraic rules can be added to the decomposition strategy to achieve even further improvements in simulation efficiency. As such, our model demonstrates the ability of an AI-agent to reason about a task that typically requires strong mathematical reasoning skills and a deep understanding of the algebraic

---

[*]Co-first authors

38th Conference on Neural Information Processing Systems (NeurIPS 2024).

structures underlying quantum circuits. Furthermore, it improves our capacity to conduct scientific research in the increasingly important field of quantum computing.

# 1 ZX-Calculus

Quantum algorithms can be expressed graphically in circuit notation, with quantum gates composed together in a time-ordered structure. The ZX-calculus [11, 12, 23, 36], offers a powerful alternative which has proven effective for reasoning about quantum computing problems such as circuit compilation and optimisation [5, 8, 13–15, 18, 27, 29] as well as classical simulation [2, 9, 10, 22, 24, 26, 33–35]. In our work we use a variation of the ZX-calculus comprised of graphs whose vertices, called *spiders*, are labelled by a real number $\in [0, 2\pi)$ (the *phase*) and two types of edges:

$$
m \left\{ \vcenter{\hbox{\includegraphics{spider}}} \right\} n \; := \; \begin{pmatrix} 1 & 0 & \dots & 0 \\ 0 & 0 & \dots & 0 \\ \vdots & \vdots & \ddots & \vdots \\ 0 & 0 & \dots & e^{i\alpha} \end{pmatrix} \updownarrow 2^n \qquad \underline{\hspace{2cm}} \; := \; \begin{pmatrix} 1 & 0 \\ 0 & 1 \end{pmatrix}
$$

$$
\text{-----} \; := \; \frac{1}{\sqrt{2}} \begin{pmatrix} 1 & 1 \\ 1 & -1 \end{pmatrix}
$$

The way spiders are wired together by edges in a ZX-diagram with $m$ inputs and $n$ outputs determines a matrix in $\mathbb{C}^{2^n \times 2^m}$. Furthermore, wiring the inputs of one diagram to the outputs of another amounts to multiplication of their respective matrices, while juxtaposing two diagrams in parallel amounts to taking the Kronecker product. Indeed, for arbitrary $m, n$ the ZX-calculus is sufficient to express any matrix in $\mathbb{C}^{2^n \times 2^m}$, hence any quantum circuit acting on qubits. In particular, standard gates in quantum computing may be expressed as ZX-diagrams:

$$
\begin{aligned}
Z &= \;\boxed{\pi}\; & T &= \;\boxed{\tfrac{\pi}{4}}\; & CNOT &= \vcenter{\hbox{\includegraphics{cnot}}} & CZ &= \vcenter{\hbox{\includegraphics{cz}}} \\
S &= \;\boxed{\tfrac{\pi}{2}}\; & H &= \;\text{-o----o-}\;
\end{aligned}
$$

Note that when no number is present on a spider, the phase is implicitly taken to equal $0$. Diagrams may be deformed arbitrarily and still represent the same quantum computation, provided the graph topology is conserved. They may also be modified using *rewrite rules*, which express how sub-diagrams may be replaced without changing the semantics (the matrix they represent) [36]:

$$
\vcenter{\hbox{\includegraphics{rw1}}}
$$

Note that $\alpha, \beta \in \mathbb{R}$, $a \in \{0, 1\}$ and addition is taken modulo $2\pi$ in the above diagrams. Rewrite rules can be used to simplify ZX representations of quantum circuits, an example of which may be found in appendix section B.1.

In the literature, there is nomenclature for certain important classes of ZX-diagram. Spiders whose phases are multiples of $\frac{\pi}{2}$ are referred to as *Clifford spiders*, hence diagrams containing only Clifford spiders are referred to as *Clifford diagrams*, also known as *stabiliser diagrams*. Spiders whose phases are an odd multiple of $\frac{\pi}{4}$ are often referred to as *T-spiders*, hence we call diagrams whose spiders only have multiple of $\frac{\pi}{4}$ phases *Clifford+T diagrams*. Moreover, we call diagrams with neither inputs nor outputs *closed diagrams*. Clifford+T diagrams are sufficient for approximating any quantum computation to arbitrary accuracy [25], but cannot be classically simulated efficiently. Clifford diagrams, on the other hand, can be classically simulated efficiently but are not universal for quantum computing. This manifests in that Clifford diagrams containing $N$ spiders may be simulated in $O(N^3)$ operations [22], whereas Clifford+T diagrams require operations exponential with the number of T-spiders. Furthermore, while the rewrite rules above are sufficient to reduce any closed Clifford ZX-diagram to a single scalar value, more advanced techniques (such as the graph decompositions described below) are required to compute the scalar of a closed Clifford+T diagram without resorting to matrix calculations.

## 2 Circuit Simulation via Graph Decompositions

Where near-term quantum hardware is insufficient for computing quantum circuits of non-trivial scale, the use of classical simulation can be very helpful in verifying their behaviour. Specifically, there is *weak* simulation, wherein a quantum circuit is emulated to provide some probabilistic output, and *strong* simulation, where the probability of a particular measurement outcome is determined. The latter is strictly more powerful as it can be used to achieve the former, and is the focus of our work.

Strong simulation of a quantum circuit can be performed by first representing it as a ZX-diagram, then reducing it to a scalar number via rewrite rules – this scalar represents the probability amplitude of the quantum computation. Where the rewrite rules are insufficient, *decompositions* may be employed to remove problematic sub-diagrams at the cost of splitting the original diagram into a weighted sum of diagrams. For Clifford+T diagrams, one state of the art decomposition used for classical simulation is the $|\mathrm{magic}_5\rangle$ decomposition, introduced by Kissinger et al. [24]:

**Lemma 1.** *The* $|\mathrm{magic}_5\rangle$ *decomposition [24]:*

*exchanges a set of* 5 *T-spiders for* 3 *partial stabiliser terms.*

This decomposition removes 4 T-spiders at the cost of replacing a single graph term with 3 terms. Efficient decompositions for Clifford+T diagrams remove hard-to-simulate T-spiders while introducing as few new terms as possible in the weighted sum. We quantify this efficiency via the *decomposition efficiency coefficient* $\alpha$, defined as follows:

**Definition 1.** *The efficiency of a particular decomposition can be measured via:*

$$\alpha := \frac{\log_2 N}{t}$$

*where* $N$ *is the number of terms produced and* $t$ *is the number of T-spiders removed by the decomposition. The overall efficiency of a sequence of decompositions and diagram rewrites,* $\alpha_{\mathrm{effective}}$, *can be measured similarly.*

A lower $\alpha$ means a more efficient decomposition. For the $|\mathrm{magic}_5\rangle$ decomposition of equation (1), the efficiency is $\alpha \approx 0.396$. In practice, the decomposition is applied to a sub-circuit, and after diagram rewrites have been applied to the resulting terms, the number of T-spiders remaining may be reduced even further. This can lead to an $\alpha_{\mathrm{effective}}$ far lower than $0.396$.

The present state-of-the-art-algorithm [24] deterministically applies efficient structure-specific decompositions (see appendix A) wherever applicable, relying on the $|\mathrm{magic}_5\rangle$ decomposition only when these structures are no longer found. In each case, this algorithm selects the 5 T-spiders upon which to apply this decomposition at random. However, we emphasise that this choice of 5 T-spiders greatly influences the effective $\alpha$ during a sequence of decompositions and diagram rewrites. As such, selecting spiders that lead to more efficient decompositions, thus yielding fewer stabiliser terms to simulate overall can significantly reduce the computational cost of strong simulation. It is this problem of selecting spiders giving more efficient decompositions that we tackle using AI.

## 3 Experiments

**Data Generation** Training, validation and test data is generated using PyZX: the Python library for quantum circuit rewriting and optimisation using the ZX-calculus [21]. We generate three different types of ZX-diagrams: 1. Clifford+T quantum circuits, 2. grid-like diagrams, and 3. random graphs generated using the $G(n, m)$ Erdös-Rényi model [16]. Generating random samples from these 3 classes of diagram requires specifying parameters determining the diagram size and phases that appear on the spiders. Specific parameter details used for generating the data can be found in the appendix section C.1.

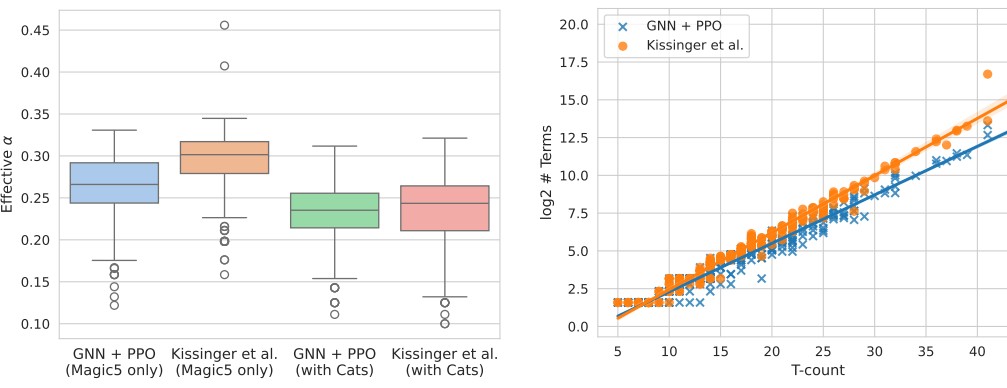

(a) Distribution of decomposition efficiency over the entire test dataset

(b) Number of stabiliser terms to simulate after decomposition against T-spider count

Figure 1: Comparison of our trained model versus Kissinger et al. [24]

**GNN Architecture** We use a graph attention network (GAT) [38]-based architecture. For the reinforcement learning algorithm used, the full architecture is divided into a features extractor, policy network and value network. The features extractor processes the input graph. The output of the features extractor is then fed into two separate networks: a policy network, which outputs a probability distribution used to sample vertices for the graph decomposition; and a value network, which assesses the relative value of the current state in the reinforcement learning environment. The value network is only used during training and is not required at inference time.

The features extractor consists of 8 GAT layers each with 4 attention heads and embedding dimension of 64. The policy and value networks follow a transformer-style architecture, consisting of blocks of alternating GAT layers with MLP layers. Residual connections, GELU activations[19], Graph normalisation layers [6], and layer normalisation [4] layers are used. We note the similarity of the policy and value network architectures to a standard transformer architecture [37], with attention layers replaced with GAT layers, and some layer normalisations replaced with graph normalisation. For further details on the architecture, see appendix section C.2.

**Reinforcement Learning Setup** We train the model using the Proximal Policy Optimisation (PPO) reinforcement learning (RL) algorithm [31] with an adapted version of generalized advantage estimation [32]. Observations in the RL environment are graphs, actions are vertices to which the $|\mathrm{magic}_5\rangle$ decomposition (1) is applied, and rewards are the effective $\alpha$-efficiencies of applying $|\mathrm{magic}_5\rangle$ to these particular vertices. Further details are given in the appendix section C.3. Data is sampled randomly during training, and intra-training performance is assessed on a validation dataset. We save model weights achieving the best performance on the validation set during a random hyperparameter search; hyperparameters for the top performing weights are listed in table 1, appendix section C.4.

**Evaluation & Results** We evaluate the models on an unseen test dataset. The best model obtains a mean effective $\alpha$ of $0.263$: a marked improvement over selecting the vertices for the decomposition randomly as in Kissinger et al. [24], which achieves $0.293$ on the same data. Note that an asymptotic decrease in $\alpha$ leads to an exponential factor speed-up. As an additional investigation, we compare efficiency coefficients when augmenting both methods with an additional set of decompositions, called the $|\mathrm{cat}_n\rangle$ decompositions (see appendix A). In both cases, the decompositions are applied according to the algorithm in Kissinger et al. [24] which is the best, to our knowledge, heuristics-based algorithm using $|\mathrm{cat}_n\rangle$ and $|\mathrm{magic}_5\rangle$. In this experiment, the model achieves a mean effective $\alpha$ of $0.232$, versus $0.235$ for [24]. This improvement in effective $\alpha$ is highlighted by figure 1a. These results are further summarised appendix D.

Moreover, when looking at the number of stabiliser terms to simulate after decomposition, our model observes better scaling behavior as the number of T-spiders, which comprise the non-stabiliser components of the ZX-diagrams, increases – see figure 1b. We hypothesise that this is because the message passing performed by the graph neural network permits, within a limited neighborhood, broader contextual information about the diagram to be taken into account when choosing the site of a decomposition, whereas the heuristic method of [24] does not.

# 4 Discussion

Our experiments have shown that a machine learning model can be perform effective mathematical reasoning, with applications to the domain of quantum computing. This was enabled by an algebraic framework: the ZX-calculus, which allowed the task of simulating quantum circuits to be formulated in terms of graphs, making the problem amenable to graph neural networks. Furthermore, the algebraic nature of the ZX-calculus provided a way of designing a reinforcement learning environment in which to learn the circuit simulation task. The trained model showed a marked improvement in simulation efficiency over existing methods without the use of AI.

These initial results are extremely promising: our methodology could be extended to include a broader set of decompositions into the model's action space. Recent work has shown that heuristics-based applications of decompositions, such as in [2, 3, 34], are remarkably effective for a broad range of quantum circuits. We also note that the ZX-calculus has applications to many other problems in quantum computing beyond circuit simulation. This suggests that similar approaches applying AI to other pertinent areas of quantum computing research, such as circuit optimisation and error correction, could be facilitated by the ZX-calculus in the same manner. Typically, these problems are solved by domain experts due to a solid understanding of the mathematics required. Our work suggests, however, that given a sufficient framework within which to perform reasoning, a machine learning model can learn to solve these mathematics-intensive problems. Indeed, it is clear that the ability of AI to solve problems requiring a high-level mathematical understanding can significantly enhance research and engineering across a broad range of scientific domains.

## Acknowledgments and Disclosure of Funding

Alexander Koziell-Pipe and Richie Yeung would like to thank Simon Harrison for his generous support via the Wolfson Harrison UKRI Quantum Foundation Scholarship, as well as the enthusiasm he shows toward their research. Alexander Koziell-Pipe and Richie Yeung are part-funded by the EPSRC.

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

## Appendix

## A   Cat state decompositions

A $|\text{cat}_n\rangle$ state is defined as follows:

$$|\text{cat}_n\rangle := \tfrac{1}{\sqrt{2}}$$

Such states can be decomposed more efficiently than 'magic' $|T\rangle^{\otimes n}$ states. In particular, the best known $|\text{cat}_3\rangle$ to $|\text{cat}_6\rangle$ decompositions are as follows [24]:

$$= \frac{e^{-i\pi/4}}{\sqrt{2}} \quad + \quad i$$

$$= \frac{e^{-i\pi/4}}{\sqrt{2}} \quad + \quad i$$

$$= \frac{1}{2} \quad + \quad \frac{ie^{i\pi/4}}{\sqrt{2}} \quad - \quad \frac{e^{i\pi/4}}{\sqrt{2}}$$

$$= \frac{1}{2} \quad + \quad \frac{ie^{i\pi/4}}{\sqrt{2}} \quad - \quad \frac{e^{i\pi/4}}{\sqrt{2}}$$

These respectively achieve decomposition efficiencies of:

$$\alpha_{|\text{cat}_3\rangle} \approx 0.333,$$
$$\alpha_{|\text{cat}_4\rangle} = 0.250,$$
$$\alpha_{|\text{cat}_5\rangle} \approx 0.317,$$
$$\alpha_{|\text{cat}_6\rangle} \approx 0.264.$$

$|\text{cat}_n\rangle$ states of larger $n$ may be decomposed into sums of the above, with an asymptotic ($n \to \infty$) efficiency equivalent to that of the $|\text{magic}_5\rangle$ decomposition, namely $\alpha \approx 0.396$ [24].

# B  ZX-Calculus Examples

## B.1  Simplification Example

The following example [36] demonstrates how a quantum circuit may be translated into a ZX-diagram and subsequently simplified via applications of the rewriting rules. The ZX representation of the same circuit is device-agnostic, highlights the symmetries of the quantum circuit, and can often be extracted to an equivalent circuit with fewer quantum gates:

## B.2  Decomposition Example

As an illustrative example, the following shows a fully reduced Clifford+T ZX-diagram and how, with the application of a single $|\mathrm{cat}_4\rangle$ decomposition, it may be reduced to two terms which are each reducible (via the rewriting rules) to a scalar:

## C  Experiment Details

### C.1  Training Data Parameters

We specify the parameters used in generating the training data below. Note that the floor function is applied to any non-integer samples drawn from the below probability distributions.

The Clifford+T circuits are generated to have a number of qubits between 20 and 30 sampled from a clipped normal distribution $\mathcal{N}(20, 25)^2$, and a qubit-dependent depth uniformly sampled from *num_qubits* $\cdot (7.5 + \mathcal{U}(0, 67.5))$.

The grid diagrams are of width sampled from a clipped normal distribution $\mathcal{N}(7.5, 1)$ between 7 and 10 nodes and a height sampled from $\mathcal{N}(5.5, 1)$ between 5 and 9 nodes.

The random graphs have a number of vertices sampled from $\mathcal{N}(25, 4)$ clipped between 20 and 32, and a number of edges sampled from $|V| \cdot (|V| - 1) \cdot (0.25 + \mathcal{U}[0, 0.15))$.

### C.2  Model Architecture

The features extractor consists of 8 GAT layers each with 4 attention heads and embedding dimension of 64. Residual connections are placed across each GAT layer, after which ReLU activations are applied. Graph normalisation layers[6] are placed between each ReLU activation and the next GAT layer.

The policy and value networks follow a transformer-style architecture, consisting of blocks of alternating GAT layers with MLP layers. Residual connections are placed across each GAT and each MLP layer, graph normalisation is placed before each GAT layer, and layer normalisation[4] is placed before each MLP layer. GELU activations are placed after each GAT and MLP layer. Note the similarity to a standard transformer architecture[37], with attention layers replaced with GAT layers, and some layer normalisations replaced with graph normalisation. Each of the policy and value network consist of 8 such GAT + MLP blocks with embedding dimension 256, only differing

---

[2]we use $\mathcal{N}(\mu, \sigma^2)$ to represent the normal distribution of mean $\mu$ and standard deviation $\sigma$ and $\mathcal{U}[a, b)$ to represent the uniform distribution sampling from the half open interval $[a, b)$.

in output: for the value network, the output vertex embeddings are aggregated into a single value by taking their mean, while the policy network outputs a single logit for each vertex in the graph.

## C.3 Reinforcement Learning Environment Description

The reinforcement learning environment begins an episode with a singleton list containing a graph to be fully decomposed. At each timestep, a graph is popped from the front of the list and the model selects vertices to decompose. A graph decomposition is applied to these vertices to produce a number of new graphs. These graphs are simplified as much as possible using classically-efficient methods. If any graph can be fully reduced to zero vertices, in other words, classically simulated efficiently, it is discarded. Otherwise, the graph is added to the list of graphs to be decomposed. This procedure is continued until no graphs remain in the list, which amounts to a classical simulation of the original graph. At each time step, a reward equal to the effective $\alpha$-efficiency of the decomposition is given.

## C.4 PPO Hyperparameters

Table 1: PPO hyperparameters used to train our model

| Hyperparameter | Value |
|---|---|
| Steps Trained | $967,680$ |
| Batch Size | $80$ |
| PPO Clip Range | $0.223$ |
| GAE gamma | $0.977$ |
| GAE lambda | $0.976$ |
| Optimizer | Adam[20] |
| Learning rate | $3 \times 10^{-4}$ |
| Entropy coefficient | $5.67 \times 10^{-3}$ |
| Value coefficient | $0.602$ |
| Min. steps per rollout | $3,840$ |
| PPO updates per rollout | $15$ |
| GAE normalisation | True |
| Gradient norm clipping | $1$ |
| Max KL divergence per rollout | $0.1$ |

# D    Detailed Evaluation Results

Table 2: Comparison between our model and [24]

| | Mean $\alpha \pm$ std dev | |
|---|---|---|
| | Magic5 | Magic5 with Cats |
| GNN + PPO | $\mathbf{0.263 \pm 0.04}$ | $\mathbf{0.232 \pm 0.04}$ |
| Kissinger et al. [24] | $0.293 \pm 0.04$ | $0.235 \pm 0.04$ |

