# OpenReview forum: "Towards Faster Quantum Circuit Simulation Using Graph Decompositions, GNNs and Reinforcement Learning"
_NeurIPS.cc/2024/Workshop/MATH-AI — MATH-AI 24_

### Official Review · Reviewer_F2EQ · 2024-10-03
**The paper presents a method to improve the simulation of quantum circuits using graph neural networks (GNNs) and reinforcement learning (RL) by optimizing circuit decompositions via the ZX-calculus framework.**

**Rating:** 5
**Confidence:** 2

**Review:**

**Summary:**
The paper presents a method to improve the simulation of quantum circuits using graph neural networks (GNNs) and reinforcement learning (RL) by optimizing circuit decompositions via the ZX-calculus framework. The authors demonstrate that their approach outperforms existing non-AI methods regarding simulation efficiency for certain quantum circuit classes.

**Pros:**
- **Problem Formulation**: Combines ZX-calculus with GNNs and RL to tackle a complex quantum simulation problem, using AI to reason about algebraic structures.

**Cons:**
- **Scalability Concerns**: The computational cost of the GNN-based approach, especially for larger circuit sizes, is not comprehensively addressed.
- **Limited Generalization**: The proposed method focuses on specific types of quantum circuits (e.g., Clifford+T circuits), and its applicability to broader classes is unclear.
- **Sparse Interpretability**: The model's decisions, though improving efficiency, are not easily interpretable in terms of the mathematical properties of the quantum circuits, limiting the insights for human experts.

**Questions:**
- Can the methodology be extended to circuits beyond Clifford+T, and if so, what modifications would be necessary?

---

### Official Review · Reviewer_3HB6 · 2024-10-06
**Review of the Paper: Towards Faster Quantum Circuit Simulation Using Graph Decompositions, GNNs, and Reinforcement Learning**

**Rating:** 9
**Confidence:** 3

**Review:**

#### 1. **Summary**
The paper presents a novel approach to simulating quantum circuits by combining the ZX-calculus, graph neural networks (GNNs), and reinforcement learning. The authors develop a method to improve the efficiency of quantum circuit simulation, particularly for circuits that cannot be classically simulated using conventional methods. By using GNN-based agents in a reinforcement learning environment, the model is trained to perform better decompositions of quantum circuits, specifically optimizing the selection of subcircuits (T-spiders) for decomposition. The proposed approach demonstrates marked improvements over existing methods, providing an efficient solution for simulating quantum circuits in the noisy intermediate-scale quantum (NISQ) era.

#### 2. **Strengths**
- **Innovative Integration**: The combination of graph neural networks and reinforcement learning to tackle the problem of quantum circuit simulation is highly innovative. The approach leverages modern AI techniques in a domain that traditionally requires strong mathematical reasoning, presenting a unique solution.

- **Use of ZX-Calculus**: The use of the ZX-calculus provides a strong mathematical framework for representing quantum circuits as graphs. This not only simplifies quantum operations but also makes the problem more amenable to AI methods like GNNs, which are naturally suited for graph-based data.

- **Performance Gains**: The paper provides compelling results that demonstrate the efficiency gains of the proposed method, with a reduction in the number of stabilizer terms to simulate and improved decomposition efficiency metrics.

- **Thorough Experimental Setup**: The experiments are well-structured, using real-world quantum circuits as well as randomly generated graph-like structures, providing a comprehensive evaluation of the model’s performance.

#### 3. **Weaknesses**
- **Limited Exploration of Practical Applications**: While the paper demonstrates promising results, there is limited discussion on how these improvements translate to practical quantum computing tasks or benchmarks. More emphasis on real-world applicability, such as how this method would be used in industry or within existing quantum computing frameworks, would strengthen the paper.

- **Complexity of GNN and RL Setup**: The reinforcement learning environment and GNN-based architecture are described in technical terms, but some parts of the explanation are too dense for non-experts in machine learning. A more intuitive explanation or visualization of the model architecture and the RL setup would help broaden the paper’s accessibility to readers from other domains, especially quantum computing researchers unfamiliar with deep learning techniques.

- **Scalability Considerations**: The paper would benefit from further discussion on how the proposed method scales as the size of the quantum circuits increases. While improvements are demonstrated for certain types of circuits, it would be valuable to discuss the upper bounds of the method’s applicability and how it compares to other state-of-the-art simulation techniques at scale.

#### 4. **Suggestions for Improvement**
- **Real-World Use Cases**: Incorporating examples of how this method could be applied to real-world quantum computing tasks or specific applications would provide a clearer picture of its potential impact. This could include more detailed comparisons with quantum circuit simulators currently used in practice.

- **Model Interpretability**: The GNN and reinforcement learning framework is quite complex. To make the approach more accessible to a wider audience, it may help to include additional interpretative explanations, visualizations, or simplified versions of the model’s inner workings.

- **Scalability and Efficiency**: Provide more detailed analyses of the scalability of the proposed method, particularly for larger circuits. This would help assess its practical feasibility for simulating larger, more complex quantum systems.

#### 5. **Conclusion**
The paper presents a highly innovative approach to simulating quantum circuits by integrating advanced AI techniques with the ZX-calculus framework. The experimental results are promising, demonstrating clear improvements over existing methods. However, more emphasis on real-world applicability, scalability, and model accessibility would enhance the paper's impact. Overall, this work represents a valuable contribution to the fields of quantum computing and machine learning, with potential applications in optimizing quantum circuit simulation and advancing quantum hardware research.

---

### Official Review · Reviewer_xxrc · 2024-10-09
**Not familiar with ZX-calculus but the experiment seem convincing**

**Rating:** 6
**Confidence:** 2

**Review:**

In this paper, the authors train a graph neural network (GNN) using reinforcement learning (RL) to simulate quantum circuits through the ZX-calculus.

The challenge of selecting optimal sub-circuit decompositions is formulated as a reinforcement learning problem.

It is recommended to include an introduction to ZX-calculus in the appendix to ensure the paper is self-contained, as the current introduction in Section 1 lacks sufficient detail, and the figures without captions are difficult to interpret.

This addition could significantly enhance the study of quantum computing algorithms. While I am not deeply familiar with quantum circuits and therefore cannot fully assess the practical impact of the contribution on quantum computing simulations, the concept of "alpha" is intriguing. Optimizing this parameter has the potential to exponentially accelerate computations.

---

### Decision · Program_Chairs · 2024-10-09

Accept